# BFS: Back-to-Front Layered Image Synthesis via Knowledge Transfer

## Abstract

Layered images have long served as a crucial representation for creative editing, and the advent of large-scale generative models has recently spurred interest in their automatic generation. Nevertheless, existing approaches remain limited. Decomposition-based methods often struggle to achieve clean separation of layers in complex scenes, while generation-based methods face challenges stemming from their training data construction pipeline, which lead to suboptimal visual quality and limited scene diversity. In this paper, we propose BFS, a novel generation-based framework for layered image synthesis. We approaches the task through the highly practical background-to-foreground formulation. Specifically, given a background layer and user guidance, it synthesizes a foreground layer that incorporates visual effects such as shadow and reflection while harmonizing with the background to form a coherent composite image. Since constructing suitable training data is difficult, we instead leverage the comparatively easy-to-learn knowledge of composite synthesis for the foreground synthesis. To this end, we design a dual-branch framework that jointly generates a composite image and a foreground layer, enabling knowledge transfer through bidirectional information exchange between the two branches. To promote this transfer, we also propose a two-stage training scheme that does not rely on ground-truth foreground layer, main dataset bottleneck. Extensive experiments show that BFS produces high-quality layered images, consistently outperforming prior methods.

## 1 Introduction

Layered images, typically organized as a stack of multiple RGBA layers, have long been a standard representation in professional image creation and editing. Such a representation enables independent manipulation of visual elements without affecting the rest of the image, and this non-destructive property not only streamlines iterative creative workflows but also enables dynamic reuse of visual elements across different compositions.

Recent advances in image generative models (Ramesh et al., 2022; Saharia et al., 2022; Rombach et al., 2022; Podell et al., 2023) have opened new opportunities for automatic layered image synthesis. According to their layer construction strategy, existing approaches can be broadly categorized into two paradigms: *decomposition-* and *generation-based* methods. *Decomposition-based* methods (Kang et al., 2025; Yang et al., 2025) extend traditional image matting approaches. They break down an existing composite image into its constituent foreground and background layers, typically guided by user-provided foreground masks. However, these methods often struggle to accurately extract foreground objects together with their associated visual effects, such as shadows and reflections, when applied to complex scenes. Moreover, they are inherently limited to separating existing images and lack the ability to synthesize new layers to construct novel compositions.

In contrast, *generation-based* methods (Zhang et al., 2023b; Huang et al., 2024; Zhang & Agrawala, 2024; Dalva et al., 2024; Huang et al., 2025b; Pu et al., 2025; Huang et al., 2025a) directly synthesize layers from random noise and text descriptions. They produce sharp and accurate alpha masks and allow the straightforward creation of new layered images using text description, which has driven growing research interest. However, since learning the distribution of layered images demands a large-scale layered image dataset, which is difficult to obtain, developing a scalable training data construction pipeline has emerged as a central challenge.

For example, Text2Layer (Zhang et al., 2023b), the MuLAn dataset (Tudosiu et al., 2024), and DreamLayer (Huang et al., 2025b) propose separation-based data construction pipelines. Starting from a composite image dataset, they extract foreground layers and then inpaint the residual regions to produce background layers. Starting from a composite image dataset, they extract foreground

layers and then inpaint the residual regions to produce background layers. However, this process often leaves visual effects in the background instead of the foreground, which in turn causes the trained model to generate incorrect outputs. Although DreamLayer (Huang et al., 2025b) filters out low-quality layered images through manual review, such reliance on human labor makes the approach unsuitable for scalable dataset construction.

Other pipelines proposed by Zhang & Agrawala (2024) and Huang et al. (2025a), start from an RGBA foreground dataset. Specifically, Zhang & Agrawala (2024) first apply outpainting outside the foreground regions to obtain composite images, then remove the foregrounds and inpaint the regions to produce background layers. While this approach allows foreground layers to retain their visual effects, these effects are typically not harmonized with the background. As a result, the generated layered images, and therefore the output trained model often exhibit poor harmonization between foreground and background layers. Moreover, the initial RGBA foreground set is limited in both diversity and scale, making the resulting composite images less representative of real-world imagery. Consequently, the generated layered images are confined to a narrow range of object categories and scene variations, limiting their applicability to broader real-world scenarios. Meanwhile, PSDiffusion (Huang et al., 2025a) hires professional designers to select assets from foreground and background datasets and carefully combine them, but this labor-intensive curation is prohibitively costly. These limitations of previous works highlight that layered image synthesis remains challenging, particularly the collecting high-quality data at scale without heavy reliance on human labor.

In this paper, we propose BFS (Back-to-Front layered image Synthesis), a novel generation-based framework for high-quality layered image synthesis. Specifically, we focus on BG2FG (background-to-foreground) synthesis, i.e., synthesizing a new foreground layer conditioned on a given background image. While alternative strategies exist such as generating all layers at once or foreground-to-background synthesis, BG2FG synthesis holds a couple of distinct practical importance. Firstly, it closely aligns with the common design process where additional objects are introduced into a scene one at a time to explore diverse layer compositions. Secondly, it naturally extends to multi-layered image generation by sequentially adding new layers.

Constructing a suitable training dataset for this task, however, is challenging. Such a dataset should be sufficiently large to provide reliable supervision for the scale, position, and pose of the foreground objects relative to the background. In addition, the foreground layers should faithfully incorporate realistic visual effects consistent with the background's contextual semantics, but collecting such foreground-background pairs remains extremely difficult.

To overcome this dataset challenge, we reduce reliance on ground-truth foreground layers and instead leverage the comparatively easier-to-learn knowledge of composite synthesis for the foreground synthesis. To this end, we first design a dual-branch generation framework that jointly synthesizes the composite image and the foreground layer, facilitating the knowledge transfer by aligning their generation pathways with bidirectional information exchange. Concretely, on a shared pretrained diffusion transformer, we attach two content LoRA modules dedicated to each modality and an information-sharing module that couples the two branches.

This dual-branch framework enables effective knowledge transfer in two respects. First, the foreground layer is generated in conjunction with the composite image rather than in isolation, allowing composition-aware synthesis that naturally incorporates realistic visual effects. Second, learning the composite branch is relatively easy, as composite generation closely aligns with the pretrained knowledge of producing high-quality images. By conditioning on the composite branch, the foreground branch can be trained stably and efficiently, even with limited data.

Furthermore, we propose a two-stage training strategy to promote knowledge transfer, without the reliance on ground-truth foreground layers. In the first stage, we learn the representations of the outputs of both branches and their compositional relationships. Specifically, we train on a synthetic dataset of well-aligned triplets of foregrounds with simulated effects, backgrounds, and composite images, using a diffusion loss and our composition loss. In the second stage, we enhance the realism of the composite branch, which naturally transfers to the foreground branch through the information-sharing learned in the previous stage. Specifically, we supervise the composite branch using a realism-rich dataset composed of backgrounds and composites, which are much easier to obtain than ground-truth foreground layers. We also introduce a regularization scheme for the foreground branch to keep its distribution during this stage.

With extensive experiments, we demonstrate that BFS achieves high-quality layered image synthesis, in which foreground and background layers are well harmonized, along with strong generalization capability compared to existing approaches. In addition, our method proves effective across diverse practical applications such as reference-based foreground generation and foreground layer extraction from composite and background images.

## 2   RELATED WORK: LAYERED IMAGE SYNTHESIS

Motivated by the practical value of layered image representations, interest in their automatic generation has grown rapidly. In this section, we review the architectural designs adopted by existing methods. Beyond early CLIP-based attempt of Text2Live (Bar-Tal et al., 2022), most contemporary methods leverage the generative power of pretrained diffusion models via fine-tuning. Decomposition-based methods (Kang et al., 2025; Yang et al., 2025) adapt the input/output layers of diffusion models to decompose a composite image into its constituent layers, thereby producing separate foreground and background layers.

Generative-based methods directly synthesize layers from random noise and text descriptions. Text2Layer (Zhang et al., 2023b) introduces a unified latent space in which foreground and background are jointly embedded, enabling both layers to be generated in a single pass. Subsequent methods (Huang et al., 2024; Zhang & Agrawala, 2024; Dalva et al., 2024; Huang et al., 2025b;a) have established a dominant paradigm of multi-pass joint synthesis, allocating a dedicated generative pathway to each layer. For instance, LayerDiff (Huang et al., 2024) synthesizes layers from layer-specific text prompts while promoting both intra- and inter-layer interactions via a collaborative attention block, and LayerDiffuse (Zhang & Agrawala, 2024) leverages layer-specific LoRA adapters for each layer and aggregates all attention vectors across all pathways to form a unified model. More recent works (Dalva et al., 2024; Huang et al., 2025b;a) augment the multi-pass design with an additional composite branch, leveraging cross-attention maps to inject global scene context into the individual layers. Despite these advances, they remain tied to U-Net–based diffusion backbones and depend heavily on cross-attention maps, with only limited exploration of emerging diffusion transformer (DiT) architectures. In contrast, the only DiT-based work, ART (Pu et al., 2025), adopts a fundamentally different strategy. It first generates a global layout, assigns latent tokens to spatial regions, and then synthesizes all layers simultaneously using a diffusion transformer. While this design enables efficient region-wise generation, it inherently limits the ability to capture holistic visual effects that span across multiple regions.

## 3   BFS (BACK-TO-FRONT LAYERED IMAGE SYNTHESIS

As introduced in Sec. 1, we tackle the BG2FG problem in layered image synthesis by proposing BFS, which is illustrated in Fig. 1. Specifically, given a background image $B$, a bounding-box mask $M$, and a foreground text description $T$, it generates a foreground RGBA layer $F = (F_{rgb}, F_\alpha)$ that contains the object specified by $T$ within $M$. The generated foreground layer is expected to (1) incorporate associated visual effects such as shadow and reflections, and (2) naturally harmonize with the background $B$, thereby producing a natural-looking composite image $C$. The composition follows the standard alpha blending equation:

$$C = F_\alpha \cdot F_{rgb} + (1 - F_\alpha) \cdot B, \tag{1}$$

where $F_{rgb}$ and $F_\alpha$ represent the RGB color components and opacity map of $F$, respectively, and $\cdot$ denotes element-wise multiplication.

The core contribution of BFS lies in transferring the relatively easier-to-learn knowledge of composite synthesis to guide more challenging foreground synthesis, thereby effectively bypassing the difficulty of constructing training data. To this end, we design a dual-branch framework that facilitates this transfer. Specifically, it simultaneously synthesizes both a composite image and a foreground layer through two dedicated branches. At the same time, it enables bidirectional information exchange between the two generation pathways, which ensures that the two modalities remain well aligned, effectively guiding the foreground branch to follow the composite branch. Architecturally, we take inspiration from UniCon (Li et al., 2024), a UNet-based joint generation framework, and adapt it to our diffusion transformer backbone.

BFS builds on Flux-Fill, an image inpainting model chosen for its task similarity. We attach two *content* LoRA modules (a composite LoRA and a foreground LoRA) to all the transformer blocks in

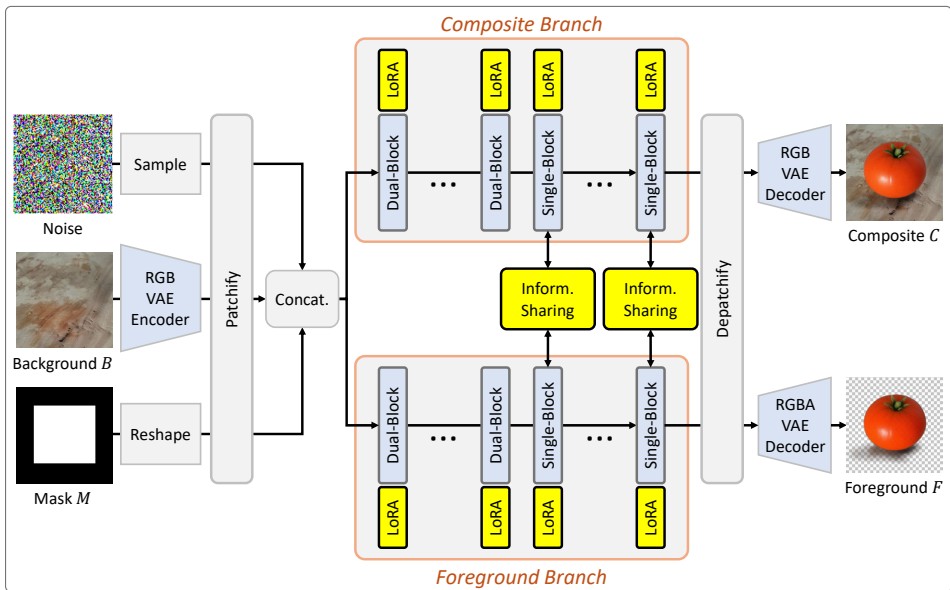

Figure 1: Overall framework of BFS. Given a background image $B$, a bounding-box $M$, and a foreground text description $T$, the model generates a foreground RGBA layer $F$. To enhance generation quality, it simultaneously synthesizes a composite image and transfers knowledge from composite generation to foreground generation. We train only the modules highlighted in yellow.

Flux-Fill. Each LoRA operates exclusively on its corresponding pathway, guiding the synthesis of each modality. To enable bidirectional information exchange between the pathways, thereby aligning both modalities, we introduce an *information-sharing* module featuring a symmetric cross-attention layer. For efficiency, this module is inserted only into the 38 single-stream blocks that follow the initial 19 dual-stream blocks.

The information-sharing module operates as follows. Let $H^C$ and $H^F$ be the intermediate features before the self-attention layer of the composite and foreground pathways at a single-stream block, respectively. We compute two cross-attentions by querying one pathway with keys/values from the other:

$$Z^{C \to F} = \text{softmax}\left(\frac{Q^F (K^C)^\top}{\sqrt{d}}\right) V^C, \qquad Z^{F \to C} = \text{softmax}\left(\frac{Q^C (K^F)^\top}{\sqrt{d}}\right) V^F, \qquad (2)$$

where $Q$, $K$, and $V$ denote the query, key, and value embeddings of each branch, and $d$ is the embedding dimension. The two cross-attention messages are concatenated, passed through a lightweight MLP $g(\cdot)$, and split into each branch via residual updates:

$$[\Delta H^C, \Delta H^F] = g(\text{concat}(Z^{F \to C}, Z^{C \to F})), \quad \widetilde{H}^C = H^C + \Delta H^C, \quad \widetilde{H}^F = H^F + \Delta H^F. \quad (3)$$

For cross-attention projection, we use shared and frozen base matrices $W_Q$, $W_K$, and $W_V$, and introduce two *direction-specific* LoRAs. The resulting feature embeddings are computed as:

$$Q^F = H^F\big(W_Q + \Delta W_Q^{C \to F}\big), \ K^C = H^C\big(W_K + \Delta W_K^{C \to F}\big), \ V^C = H^C\big(W_V + \Delta W_V^{C \to F}\big),$$

$$Q^C = H^C\big(W_Q + \Delta W_Q^{F \to C}\big), \ K^F = H^F\big(W_K + \Delta W_K^{F \to C}\big), \ V^F = H^F\big(W_V + \Delta W_V^{F \to C}\big).$$
$$(4)$$

The base projections $(W_Q, W_K, W_V)$ are initialized from the pretrained self-attention weights of each block. The final trainable components of BFS are the two *content* LoRAs, the two *direction-specific information-sharing* LoRAs $\{\Delta W_{\{\cdot\}}^{C \to F}, \Delta W_{\{\cdot\}}^{F \to C}\}$, and the lightweight MLP.

At inference time, the diffusion denoising process operates in the latent space of the pretrained VAE. Two same random noise tensors are first channel-wise concatenated with the conditioning inputs (the background latent and a bounding box mask reshaped following Flux-Fill). The concatenated tensors are then stacked along the batch dimension and passed through the diffusion transformer with text embeddings computed from foreground descriptions. The two content LoRAs are applied independently to different batch elements, while the information-sharing module attends jointly to both.

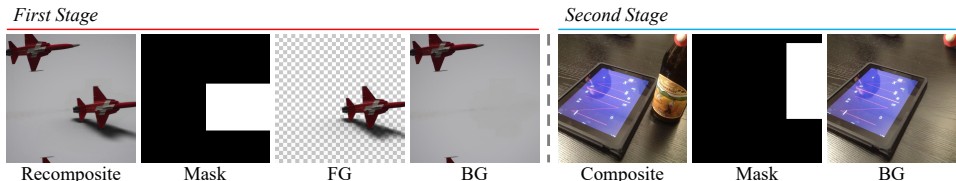

First Stage | Second Stage

Recomposite    Mask    FG    BG    Composite    Mask    BG

Figure 2: Training dataset samples used in each training stage.

After iterative denoising, the model produces a composite latent and a foreground latent. Finally, the only foreground latent is decoded to image domain using RGBA VAE introduced in LayerDiffuse (Zhang & Agrawala, 2024). Further implementation details are provided in the appendix.

## 4 TWO-STAGE TRAINING STRATEGY

To train BFS, the most straightforward dataset would comprise foreground, background, composite images, and bounding-box masks. Such a dataset must be sufficiently large to provide reliable supervision for the scale, position, and pose of the foreground object, while the foreground layers should be represented in RGBA format and faithfully capture realistic visual effects. Yet, as discussed in Sec. 1, constructing a layered image dataset in practice is highly challenging. Although background and composite image pairs can be obtained relatively easily through recent object removal techniques, acquiring corresponding foreground representations remains extremely difficult. To address this challenge, we propose a two-stage training strategy that avoids reliance on explicit ground-truth foreground layers. In brief, we first train our model on a simulation-based synthetic dataset containing all modalities, and then fine-tune it on realism-rich background and composite pairs to enhance visual fidelity and contextual harmonization of the composites, which in turn improves the quality of the foreground layers through the information-sharing module. The following paragraphs provide details of the datasets and the training losses used in each stage.

**Representation Learning with Compositional Consistency**    BFS first learns modality-specific representations and their compositional relationships using a simulation-based synthetic dataset composed of triplets of foreground, background, composite images, along with bounding-box masks. To construct the dataset, we adopt an approach similar to LayerDecomp (Yang et al., 2025) on the RORem object removal dataset (Li et al., 2025), which provides composite images, object masks, and background images where both objects and their visual effects have been removed. For each composite image $C$, we first extract a foreground layer $F$ from the object mask $M$ using a matting technique (Yao et al., 2024), then augment it to include simulated shadows using an internal shadow generation network. The augmented foreground layer $\hat{F}$ is recomposited with the background layer $B$ to form a new composite image $\hat{C}$. Finally, we replace the original object mask with a bounding-box mask $\hat{M}$, resulting in a training corpus of $\{\hat{F}, B, \hat{M}, \hat{C}\}$, as shown in the left side in Fig. 2. We also provide visual examples of the dataset construction pipeline in the appendix. For notational simplicity, we drop the hat in the rest of the paper.

Our data construction pipeline produces image layers with precise pixel-level alignment and embeds simulated visual effect into the foreground layer, even though these effects are less realistic and not harmonized with the background layer. In contrast to LayerDecomp, which constructs composites by randomly pairing foregrounds with backgrounds, our method derives foregrounds directly from real composite images, thereby offering more reliable supervision for object placement and scale.

Using the synthetic dataset, we employ a flow-matching loss. Let $z_F, z_B, z_C$ denote the VAE latents of the foreground, background, and composite images, respectively. At each training iteration, we sample Gaussian noise $z_0 \sim \mathcal{N}(0, I)$ and linearly interpolate it with each data latent $z_1 \in \{z_F, z_C\}$ at a random time step $t \sim \mathcal{U}(0, 1)$: $z^{(t)} = (1 - t)z_0 + tz_1$ The flow-matching objective is then defined as:

$$\mathcal{L}_{\text{flow}} = \mathbb{E}_{t, z_0, z_1}\left[\left\|v_C^* - v_\theta^C(z_C^{(t)}, z_F^{(t)}, t, \kappa)\right\|_2^2 + \left\|v_F^* - v_\theta^F(z_C^{(t)}, z_F^{(t)}, t, \kappa)\right\|_2^2\right], \tag{5}$$

where $v_C^*$ and $v_F^*$ are the ground-truth velocity fields (given by $v^* = z_1 - z_0$). $v_\theta^C$ and $v_\theta^F$ are velocity fields predicted by the composite and foreground branches, respectively. Note that $v_\theta^C$ and $v_\theta^F$ are computed from both $z_C^{(t)}$ and $z_F^{(t)}$, since the composite and foreground branches are connected through information-sharing modules. Here, $\kappa$ represents the conditioning inputs, including the foreground text description $T$, the background latent $z_B$, and a down-sampled mask $M$.

To further enforce compositional consistency between the branches, we introduce a composition loss, which is defined as:

$$\mathcal{L}_{\text{comp}} = \left\| \Phi(\hat{z}_F, z_B) - \hat{z}_C \right\|_1, \tag{6}$$

where $\hat{z}_C$ and $\hat{z}_F$ are the model's one-step denoised estimates at time $t$, (i.e., the predicted clean latents from $z_C^{(t)}$ and $z_F^{(t)}$). $\Phi$ is a function that computes a composite of $\hat{z}_F$ and $z_B$ in the latent space. $\Phi$ is modeled as a neural network with learnable parameters and trained in advance to train our framework. Finally, the overall objective of the first stage is defined as:

$$\mathcal{L}_{\text{first}} = \mathcal{L}_{\text{flow}} + 0.1\mathcal{L}_{\text{comp}}. \tag{7}$$

Details of the auxiliary networks used during training are provided in the appendix.

**Realism Enhancement with Distribution Regularization**  In this stage, we enhance the realism of the generated images using a realism-rich dataset that captures diverse real-world visual effects. Specifically, we utilize paired background and composite images to supervise only the composite LoRA so that the generated composite latent better matches realistic image statistics. The resulting improvements then propagate to the foreground branch through the information-sharing modules, aligning the foreground with the refined composite and calibrating its visual effects.

For the second stage, we construct a dataset based on the object-removal dataset RORem (Li et al., 2025), which provides background and composite images and bounding-box masks. The background images of RORem are generated by inpainting the masked regions in the composite images, but they often retain residual visual effects of foreground objects such as shadows and reflections. To obtain cleaner backgrounds, we instead synthesize background images using ObjectClear (Zhao et al., 2025), a more advanced object-removal method that removes both objects and their associated visual effects. This dataset construction pipeline is scalable and readily allows further expansion of the training corpus in the future.

Based on the dataset, we define a realism-boosting loss that supervises only the composite LoRA:

$$\mathcal{L}_{\text{real}} = \mathbb{E}_t \left[ \left\| v_C^* - v_\theta^C(z_C^{(t)}, z_{\tilde{F}}^{(t)}, t, \kappa) \right\|_2^2 \right], \tag{8}$$

where $v^* = z_1 - z_0$. As no ground-truth foreground layer $F$ is available, we instead use a *surrogate* input for the foreground branch, specifically $\tilde{F} = C \cdot M$, obtained by masking the composite image $C$ with the binary mask $M$. We empirically find that this surrogate input has negligible influence on learning dynamics.

While the composite supervision improves realism, relying on it alone may cause the foreground distribution to drift away from the distribution established in the previous stage, degrading the quality of the foreground layer. To mitigate this, we introduce additional supervision for the foreground LoRA and the information-sharing module using the synthetic dataset of the previous stage. Although the dataset is not fully realistic, it benefits from the pixel-wise alignment across $\{F, B, C\}$. Specifically, when we feed the clean composite latent (without adding noise) as the input of the composite branch with timestep $t = 0$, the branch is expected to faithfully reconstruct the clean composite latent. At the same time, the foreground branch should estimate the residual between $C$ and $B$ (i.e., $F$) to satisfy the compositional relationship. To encourage this behavior, we introduce a regularization loss:

$$\mathcal{L}_{\text{reg}} = \mathbb{E}_t \left[ \left\| v_F^* - v_\theta^F(z_C, z_F^{(t)}, (0, t), \kappa) \right\|_2^2 \right] \tag{9}$$

where $(0, t)$ indicates that we use different timesteps for the composite and foreground branches, i.e., we set the timestep to 0 for the composite branch, while setting it to $t$ for the foreground branch. During training iterations, we alternate between optimizing the realism-boosting loss and the foreground regularization loss, each combined with the composition loss. Specifically, we optimize $\mathcal{L}_{\text{real}}$ and $\mathcal{L}_{\text{reg}} + 0.1\mathcal{L}_{\text{comp}}$ in turn.

## 5 EXPERIMENTS

### 5.1 COMPARISON WITH OTHER LAYERED IMAGE SYNTHESIS METHODS

**Baselines**  We compare the quality of the layered images generated by BFS and existing approaches on multiple datasets, including SAM-FB (He et al., 2024), RORem (Li et al., 2025), and AnyInsertion (Song et al., 2025). Each dataset provides background images and masks specifying the regions

Table 1: Quantitative evaluation of existing layered image synthesis methods on SAM-FB dataset.

| Method | FG | Recomposite (or Background for LayeringDiff + ObjectClear) | | | |
|---|---|---|---|---|---|
| | CLIP (↑) | MUSIQ (↑) | MANIQA (↑) | KID×1000 (↓) | DINO (↑) |
| LayeringDiff + ObjectClear | 0.59 | 70.33 | 0.430 | 0.72 | **1.00** |
| LayerDiffuse | **0.78** | 70.47 | 0.411 | 10.89 | 0.50 |
| Ours | 0.63 | **70.99** | **0.432** | **0.62** | 0.96 |

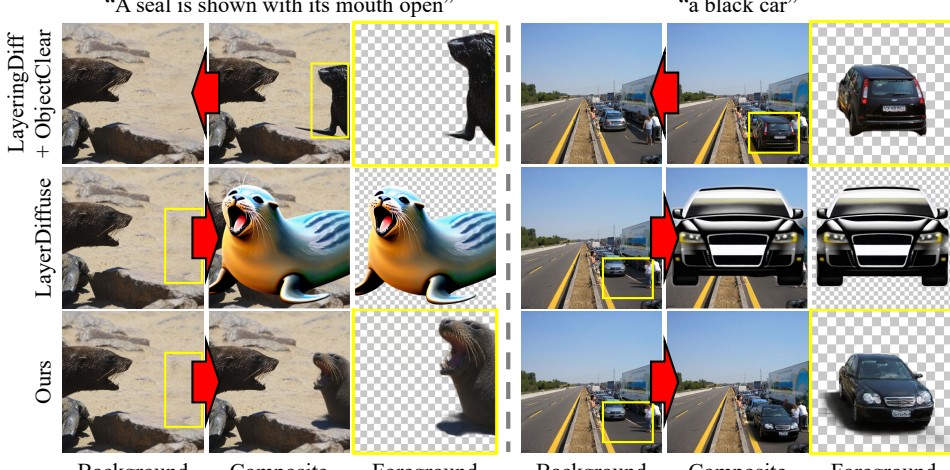

Figure 3: Qualitative comparison with existing layered image synthesis approaches. For better visual comparison, we present zoom-in views of the generated foreground layer except for LayerDiffuse, as its foreground spans the entire region.

where foreground objects should be placed. For consistency, all masks are converted into bounding-box form. All input images (backgrounds and masks) are resized such that the shorter side is 512 pixels before being fed into the models. For the target foreground text descriptions, we use the captions provided by the SAM-FB and AnyInsertion datasets, while those for the RORem dataset are generated using BLIP2 (Li et al., 2023).

As the decomposition-based methods cannot generate new foreground layers, we instead use them to decompose the ground-truth composite images into layered images and evaluate them. Specifically, we compute foreground masks using SAM (Ravi et al., 2024) based on the bounding boxes of the object masks, and employ LayeringDiff (Kang et al., 2025) to estimate the foreground layers. Background layers are generated with ObjectClear (Zhao et al., 2025), since LayeringDiff often leaves residual visual effects in the background. For LayeringDiff, we use the code provided by the authors, while for ObjectClear we use the official implementation. LayerDecomp (Yang et al., 2025) could not be evaluated because its code is not publicly available, but the quality of its generated background layers can be reasonably assumed to be comparable to that of ObjectClear, given the close similarity in their training strategies.

For generation-based methods, we compare only with LayerDiffuse (Zhang & Agrawala, 2024), as it is the only method with publicly available code. Specifically, we adopt its strongest SDXL-based two-stage variant, which first generates a composite image and then estimates the foreground layer conditioned on the background and the composite.

**Qualitative Comparison** Fig. 3 presents a qualitative comparison. For clearer visualization, we enlarge the foreground regions generated by our method and by LayeringDiff combined with ObjectClear. As shown in the figure, LayeringDiff fails to disentangle fine details in complex scenes, such as the whiskers of a seal or the side mirror of a car. The generated foreground objects by LayerDiffuse are frequently oversized and inconsistent with the background context, leading to poor visual harmony. This indicates that the synthetic dataset used for training does not generalize well to real-world backgrounds, leaving the model uncertain about how to position objects, and simply producing large, centrally placed ones that follow the typical distribution of RGBA images. In contrast, our method generates foreground objects that are contextually appropriate, visually consistent, and equipped with proper visual effects, even in complex scenes, yielding more coherent composites.

**Quantitative Comparison** Quantitative evaluation of generated layered images is inherently challenging, as ground-truth layered images are not available. In this work, we evaluate both the gener-

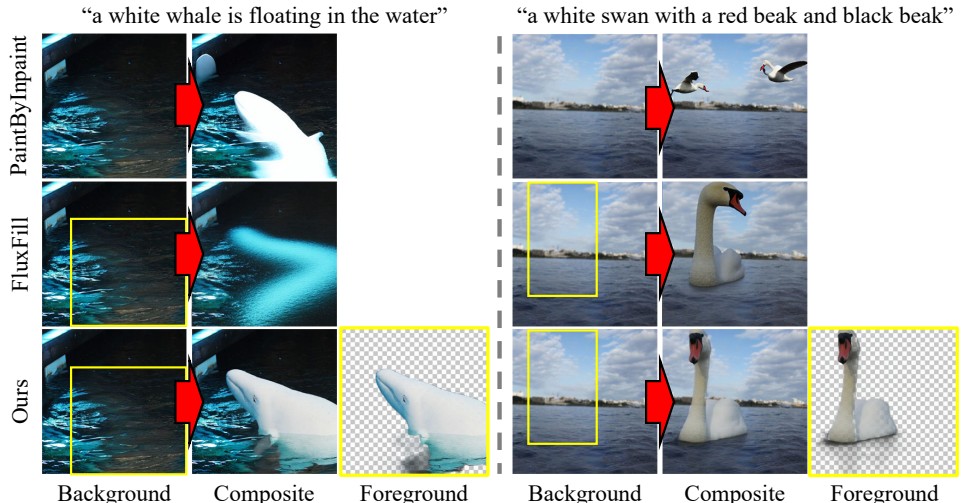

Figure 4: Qualitative comparison with an object insertion method PaintByInpaint (Wasserman et al., 2025) and a image inpainting method FluxFill (Labs, 2024)

Table 2: Quantitative evaluation with an object insertion method FluxFill and an image inpainting method PaintByInpaint on SAM-FB datset.

| Method | MUSIQ (↑) | MANIQA (↑) | KID×1000 (↓) | DINO (↑) |
|---|---|---|---|---|
| PaintByInpaint (Wasserman et al., 2025) | 70.89 | 0.41 | 0.76 | 0.93 |
| FluxFill (Labs, 2024) | 70.71 | 0.42 | 0.79 | **0.96** |
| Ours | **70.99** | **0.43** | **0.62** | **0.96** |

ated foreground layer and the recomposite image obtained by combining the generated foreground with the input background. We report results on the SAM-FB dataset, while results on two additional datasets are provided in the appendix. For decomposition-based methods, where the input is a composite image, we instead evaluate the estimated background layers rather than the composite image. For evaluation metrics, the quality of the foreground layers is assessed using the CLIP score (Hessel et al., 2021), which measures the alignment between the generated (or decomposed) foreground and the input foreground caption. For recomposite evaluation, we employ non-reference aesthetic quality metrics, MUSIQ (Yu et al., 2025), MANIQA (Yu et al., 2025), as well as FID (Heusel et al., 2017) to measure distributional similarity against the target distribution. In addition, we report the DINO score (Oquab et al., 2023) to directly compare with ground-truth composite images. The results are summarized in Tab. 1. LayerDiffuse attains the highest CLIP scores, as it often generates large foreground objects that dominate the image and are therefore favored by the metrics. However, its performance on recomposite evaluation is notably worse than other methods, especially in KID and DINO, where the generated results deviate substantially from the ground-truth distribution. Our method achieves a higher CLIP score than LayeringDiff, while also producing foreground layers that yield recomposite images with superior aesthetic quality and closer alignment to the ground-truth.

## 5.2 COMPARATIVE WITH OTHER OBJECT INSERTION METHODS

BFS is conceptually related to naturally adding new objects into a given scene. To evaluate this capability, we compare it against two recent models designed for related tasks: the image inpainting model Flux-Fill (Labs, 2024) and the object insertion method PaintByInpaint (Wasserman et al., 2025). For Flux-Fill, we adopt the default configuration. Since PaintByInpaint does not take an input mask and follows an instruction-based text description, we prepend the word "add" to the foreground caption before providing it to the model.

Fig. 4 presents a qualitative comparison of the composites generated by these baselines and recomposite image by our approach. For reference, we also show the foreground layers produced by BFS. As shown in the figure, Our method synthesizes high-quality recomposite images comparable to existing approaches, while additionally generating explicit and reusable foreground layers that facilitate further editing and composition. Tab. 2 summarizes the quantitative evaluation on the SAM-FB dataset. The results show that our method achieves slightly higher scores than existing approaches, demonstrating that BFS produces highly natural recomposite images comparable to those of state-of-the-art insertion models trained solely with real-world supervision.

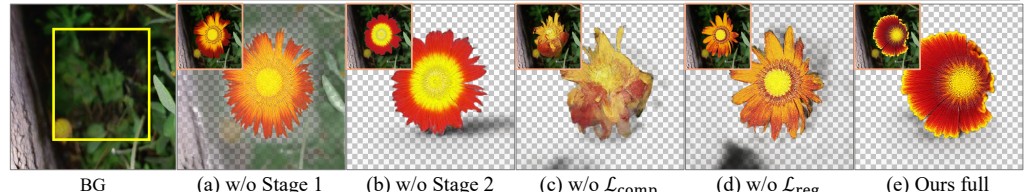

*"A yellow and red flow with a yellow center"*

| BG | (a) w/o Stage 1 | (b) w/o Stage 2 | (c) w/o $\mathcal{L}_{comp}$ | (d) w/o $\mathcal{L}_{reg}$ | (e) Ours full |

Figure 5: Ablation study. Qualitative comparison of BFS with different components removed. The inset in the top-left shows the output of the composite branch for reference.

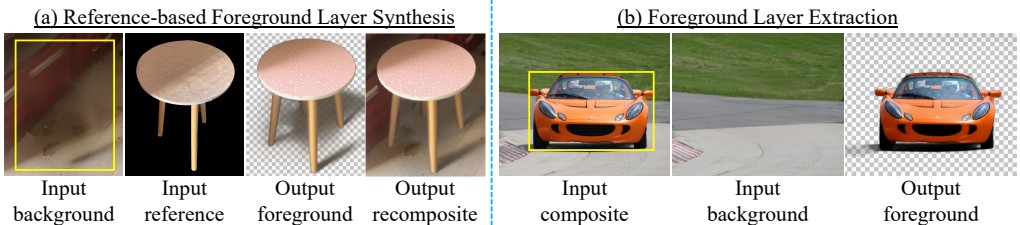

| (a) Reference-based Foreground Layer Synthesis | | | | (b) Foreground Layer Extraction | | |

| Input background | Input reference | Output foreground | Output recomposite | Input composite | Input background | Output foreground |

Figure 6: Practical applications of BFS. (a) Reference-based foreground layer synthesis guided by a reference image. (b) Foreground layer extraction from a composite and its background.

## 5.3 ABLATION STUDY

We conduct an extensive ablation study to evaluate the contribution of each component in BFS. Fig. 5 presents qualitative comparisons of the results obtained by different model variants, given a background image and a bounding-box mask (highlighted in yellow). We also include the outputs of the composite branch for each variant in the figure inset. Without Stage 1 representation learning, directly training Stage 2 completely fails to capture the foreground distribution. In contrast, omitting Stage 2 synthesize foreground layers that lack realism showing unnatural shadow effect. Removing the composition loss in Stage 1 severely degrades both branches, as effective knowledge transfer does not occur, which diminishes the quality of the foreground layers and the low-quality information is passed to the composite branch, which consequently harms the composite branch. Excluding the regularization step in Stage 2 causes the RGBA distribution of the foreground layer to collapse, introducing irrelevant visual effects into the representation. By contrast, our full model successfully aligns the two branches and produces high-quality RGBA layers.

## 5.4 FURTHER APPLICATIONS

We present practical applications of BFS. It enables *reference-based foreground synthesis*: by substituting the caption embedding with that of a reference image via a Flux-Redux adapter, the synthesized foreground inherits the reference characteristics (Fig. 6 (a)). BFS also supports *foreground extraction* from existing images. Given a composite and its background, we input the composite into the composite branch (instead of random noise) with its timestep fixed to zero, yielding a residual layer corresponding to the foreground. Unlike conventional matting, this produces a layer that preserves both the object and realistic visual effects (Fig. 6 (b)).

## 6 CONCLUSION

We present BFS, a generation-based framework for high-quality layered image synthesis. To overcome the difficulty of constructing a suitable training dataset, we transfers the easier-to-learn knowledge of composite synthesis to foreground synthesis, by using a dual-branch framework that jointly generates composite images and foreground layers, with bidirectional information exchange aligning the two branches. To further promote effective knowledge transfer, we also propose two-stage training scheme avoids reliance on ground-truth foreground supervision, which is the main dataset bottleneck. Through extensive experiments, we demonstrated that BFS produces high-quality layered images with strong generalization ability, consistently surpassing prior methods.

BFS has certain limitations. The dual-branch design increases inference time, as generating a single output with BFS takes 31 seconds compared to 12 seconds with Flux-Fill backbone. Furthermore, the visual effects are not guaranteed to be physically correct. We will integrate rendering toolchains to capture complex, physically valid effects and use them to supervise future models.

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

# A  APPENDIX

In this appendix, we provide:

## A.1  RELATED WORK: OBJECT INSERTION

Another closely related line of research is object insertion, which aims to integrate an object into a target scene while ensuring object-level plausibility (pose, scale, and identity) and scene-level consistency (lighting, shadows, reflections, and other visual effects). Recently, large-scale image generative models have motivated *single framework solutions* that address all the subproblems simultaneously, directly producing composite images in a single synthesis. Among them, training-free methods (Lu et al., 2023b; Tewel et al., 2024) demonstrate that high-quality insertion results can be achieved by guiding the diffusion model generation process. A more mainstream direction is to fine-tune diffusion models to insert a reference image into a given background. Early studies (Yang et al., 2023; Song et al., 2023) replace text embeddings with embeddings of a given reference image to reproduce the target instance, while subsequent methods (Zhang et al., 2023a; Chen et al., 2024; Song et al., 2024) place greater emphasis on identity preservation by devising more effective identity injection strategies.

More recent researches (Huang et al., 2025c; He et al., 2024; Song et al., 2025; Canet Tarrés et al., 2024; Liang et al., 2024; Wang et al., 2025; Winter et al., 2024a; Canberk et al., 2024; Yu et al., 2025) have shifted their focus toward enhancing the realism of generated composites. To this end, some works (Huang et al., 2025c; He et al., 2024; Song et al., 2025; Canet Tarrés et al., 2024) curate dataset that explicitly capture object–scene interactions. Others, such as ObjectDrop (Winter et al., 2024a), EraseDraw (Canberk et al., 2024), ORIDa (Kim et al., 2025), and Omnipaint (Yu et al., 2025), leverage object removal data to strengthen the fidelity of inserted objects. Meanwhile, Multitwine (Tarrés et al., 2025) extends the task to multi-object insertion, and ObjectMate (Winter et al., 2024b) and DreamCom (Lu et al., 2023a) leverage multiple reference images to utilize richer view-dependent information. Finally, layout estimation and then inpaint approaches, such as SmartMask (Singh et al., 2024) and Generative Location Modeling (Yun et al., 2024), first estimate insertion layouts and exploit pretrained inpainting models to generate natural composite results.

Despite these efforts, most object insertion methods concentrate on composite image generation rather than learning disentangled representations. As a result, the inserted objects cannot be easily separated, reused, or independently edited after synthesis, limiting post-hoc editing flexibility. In contrast, our approach explicitly generates a foreground RGBA layer preserving contextual realism while providing an editable and reusable representation of the inserted object.

## A.2  IMPLEMENTATION DETAILS

For inference, we use the flow-matching Euler discrete sampler with 50 steps and a guidance scale of 30. We summarize the training configurations for both training stages of BFS in Tab. A1.

## A.3  SIMULATION-BASED SYNTHETIC DATASET CONSTRUCTION PIPELINE

We describe in detail the pipeline for constructing our simulation-based synthetic dataset used in the training Stage 1, with visual examples in Fig. A1. We begin with the RORem (Li et al., 2025) dataset, an object removal dataset that contains a composite image $C$, a rough binary mask $M$ of an object in the composite, and a background image $B$ where the object has been removed together with its visual effects.

Table A1: Training setup for Stage 1 and Stage 2 of BFS.

| Item | Stage 1 | Stage 2 |
|---|---|---|
| Datasets | Synthetic dataset from the RORem dataset | RORem dataset |
| Number of Samples | 57,304 | 114,352 |
| Image Resolution | $512 \times 512$ | $512 \times 512$ |
| RANK | 64 | 64 |
| Optimizer | AdamW 8bit ($\beta_1 = 0.9, \beta_2 = 0.999$) | AdamW 8bit ($\beta_1 = 0.9, \beta_2 = 0.999$) |
| Learning Rate | $8e - 6$ | $8e - 6$ |
| Weight Decay | $1e - 2$ | $1e - 2$ |
| Batch Size | 32 | 32 |
| Training Steps | $3,000$ | $9,000$ |
| LR Schedule | Cosine with restarts | Cosine with restarts |
| Gradient Clipping | 1.0 | 1.0 |
| Hardware | 4×A100 80GB | 4×A100 80GB |
| Precision | BF16 | BF16 |

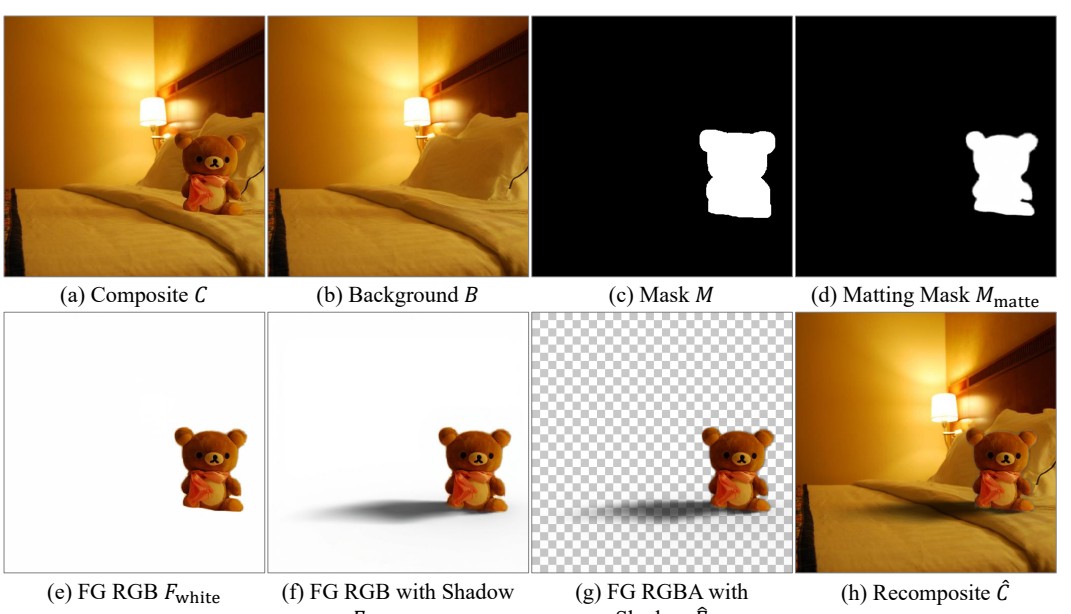

(a) Composite $C$    (b) Background $B$    (c) Mask $M$    (d) Matting Mask $M_{\text{matte}}$

(e) FG RGB $F_{\text{white}}$    (f) FG RGB with Shadow $F_{\text{shadowed}}$    (g) FG RGBA with Shadow $\hat{F}$    (h) Recomposite $\hat{C}$

Figure A1: Visual examples from the pipeline used to construct the synthetic dataset.

Starting from the rough mask $M$, we first convert it into a bounding-box form $M_{\text{bbox}}$ and apply SAM2 (Ravi et al., 2024) to detect an object mask $M_{\text{sam}}$ based on the bounding-box mask $M_{\text{bbox}}$. Since SAM2 may capture regions misaligned with the original rough mask $M$, we filter out cases where the intersection-over-union (IoU) between the two masks ($M$ and $M_{\text{sam}}$) is below 0.8, resulting in 57,304 valid samples out of the initial 114,352 images. For the retained masks, we perform an 11-pixel morphological dilation and erosion operations to construct a trimap, where pixels outside the dilated region are set to 0, those inside the eroded region to 1, and the intermediate band to 0.5. Based on the trimap, we estimate a detailed fine mask $M_{\text{matte}}$ using a image matting method, ViTMatte (Yao et al., 2024). Based on the matting mask $M_{\text{matte}}$, we extract the foreground object from the composite $C$ and place it on a white background ($F_{\text{white}}$). We then apply a shadow generation network $G_{\text{shadow}}$ to synthesize shadows cast by the object, producing a shadow-augmented foreground $F_{\text{shadowed}}$ (the details of $G_{\text{shadow}}$ are provided in Sec. A.4).

Lastly, we compute the alpha channel to form the final RGBA representation $\hat{F}$ from the shadow-augmented foreground $F_{\text{shadowed}}$. Specifically, we assume that the synthesized shadow can be represented as the product of the shadow opacity $\alpha_{\text{shadow}}$ and a pure black image $I_{\text{black}}$. Accordingly,

$F_{\text{shadowed}}$ can be expressed as

$$F_{\text{shadowed}} = \alpha_F F + \alpha_{\text{shadow}} I_{\text{black}} + (1 - \alpha_F - \alpha_{\text{shadow}}) I_{\text{white}} \tag{10}$$

where $F$ denotes the foreground object, $\alpha_F$ is the object opacity, and $I_{\text{white}}$ is the pure white background. Substituting $I_{\text{black}} = 0$ and $I_{\text{white}} = 1$ simplifies the equation to

$$F_{\text{shadowed}} = \alpha_F F + 1 - \alpha_F - \alpha_{\text{shadow}}, \tag{11}$$

which can be rearranged as

$$\alpha_F + \alpha_{\text{shadow}} = \alpha_F F + 1 - F_{\text{shadowed}}. \tag{12}$$

The left-hand side represents the combined opacity of the object and its shadow, which we denote as $\alpha$. The right-hand side is fully computable, since $\alpha_F F$ corresponds to $F_{\text{black}}$, i.e., the object placed on a black background. Based on this, we can derive the foreground RGBA representation with the computed alpha $\alpha$ as

$$\hat{F} = (\hat{F}_{\text{color}}, \alpha), \tag{13}$$

where the color component is obtained by normalizing the shadowed foreground with its opacity,

$$\hat{F}_{\text{color}} = \frac{F_{\text{shadowed}} - (1 - \alpha)}{\alpha + \epsilon}, \tag{14}$$

and $\epsilon$ is a small constant to avoid division by zero. Finally, the recomposite image can be produced by alpha blending the RGBA foreground with the background $B$:

$$\hat{C} = \alpha \cdot \hat{F}_{\text{color}} + (1 - \alpha) \cdot B. \tag{15}$$

## A.4 SHADOW GENERATION NETWORK

We introduce our shadow generation network $G_{\text{shadow}}$, which takes as input an object composited on a white background together with a fine-grained object mask, and outputs a shadow-casted image. Since no publicly available model and dataset align with our target scenario, we train our own network using a two-stage fine-tuning strategy.

In Stage 1, the model is trained to generate shadows that harmonize with the surrounding scene, given an input image and a fine-grained object mask. For this purpose, we leverage the large-scale DESOBAv2 dataset (Liu et al., 2024), which consists of $28,573$ triplets containing (i) an object copy-pasted onto a background without shadows, (ii) the corresponding scene with naturally cast shadows, and (iii) a fine-grained object mask. We fine-tune the Flux backbone (Labs, 2024) by attaching a LoRA adapter, modifying the input layer to jointly process the image and mask, and optimizing the LoRA parameters with an $\ell_2$ loss between the model outputs and the ground-truth shadowed images.

Although the Stage 1 model can produce realistic shadows, its outputs are not directly aligned with our target scenario, where objects are placed on a uniform white background. To address this mismatch, Stage 2 performs additional fine-tuning on a smaller dataset (Tasar et al., 2024) specifically tailored to white-background scenes. This dataset is constructed using Blender and provides 555 pairs of (i) an object composited on a white background with rendered shadows and (ii) a fine-grained object mask. From these pairs, we construct training triplets by extracting the object using the mask and placing it onto a white background. The network is trained with the same $\ell_2$ loss as in Stage 1. This refinement enables the model to generate shadows consistent with objects placed on a white background.

Through this two-stage training, we obtain a robust shadow generation network, which serves as a crucial component in constructing the simulation-based dataset used in our synthetic dataset construction pipeline. We summarize the training configurations for both stages in Tab. A2.

## A.5 LATENT COMPOSITION NETWORK

In this section, we introduce the latent composition network $\Phi$ used in the training Stage 1. Learning the compositional relationship where the output of the composite branch should equal the composition of the input background latent and the output of the foreground branch is crucial, as also demonstrated in the ablation study of the main paper.

Table A2: Training setup for Stage 1 and Stage 2 of the shadow generation network $G_{\text{shadow}}$.

| Item | Stage 1 | Stage 2 |
|---|---|---|
| Datasets | DESOVAv2 (Liu et al., 2024) | Tasar et al. (2024) |
| Number of Samples | 28,573 | 555 |
| Image Resolution | $512 \times 512$ | $512 \times 512$ |
| RANK | 128 | 128 |
| Optimizer | AdamW 8bit ($\beta_1 = 0.9$, $\beta_2 = 0.999$) | AdamW 8bit ($\beta_1 = 0.9$, $\beta_2 = 0.999$) |
| Learning Rate | $3e - 5$ | $3e - 5$ |
| Weight Decay | $1e - 2$ | $1e - 2$ |
| Batch Size | 32 | 32 |
| Training Steps | $22,000$ | $2,000$ |
| LR Schedule | Cosine with restarts | Cosine with restarts |
| Gradient Clipping | 1.0 | 1.0 |
| Hardware | $1\times$A100 80GB | $1\times$A100 80GB |
| Precision | BF16 | BF16 |

However, since both branches operate in the latent space of the pretrained VAE rather than directly in the image domain, the standard alpha blending equation cannot be applied. Furthermore, the encoding and decoding of the foreground latent rely on the RGBA VAE introduced in LayerDiffuse (Zhang & Agrawala, 2024), which projects transparency information into the latent representation, meaning that no explicit blending mask is available. While one could follow the approach of LayerDecomp (Yang et al., 2025) and design a composition loss in the image domain using the VAE decodings of the generated latents of both branches, this requires backpropagation through the VAE decoder. Such a strategy is computationally expensive and carries the risk of inaccurate gradient propagation due to the long backpropagation path.

To address this, we introduce the latent composition network $\Phi$, which performs composition directly in the latent space. This network adopts a simple U-Net architecture and is trained to map the channel-wise concatenation of a foreground latent and a background latent to the corresponding composite latent. To this end, the training objective is defined as the reconstruction error between the predicted composite latent $\hat{z}_C$ and the ground-truth composite latent $z_C$:

$$\mathcal{L}_{\text{comp}} = \|z_C - \Phi(z_B, z_F)\|_2^2. \tag{16}$$

Below, we provide the code for constructing the network architecture using the Diffusers library, along with details of the training setup (Tab. A3).

```python
from diffusers import UNet2DModel
composition_net = UNet2DModel(
    sample_size=128,       # latent resolution
    in_channels=32,        # foreground + background latents channels
    out_channels=16,       # composite latent channels
    layers_per_block=2,
    block_out_channels=[320, 640],
    down_block_types=("DownBlock2D", "DownBlock2D"),
    up_block_types=("UpBlock2D", "UpBlock2D"),
)
```

## A.6  EXTENDED EXPERIMENTAL RESULTS

We present additional qualitative comparison examples with layered image synthesis methods in Fig. A2 and Fig. A3.

We present additional qualitative comparison examples with inpainting methods in Fig. A4, Fig. A5, Fig. A6, Fig. A7, and Fig. A8.

We present additional quantitative comparisons on the RORem dataset (Li et al., 2025), comparing existing layered image synthesis methods in Tab. A4, and object insertion methods in Tab. A5.

We present additional quantitative comparisons on the AnyInsertion dataset (Song et al., 2025), comparing existing layered image synthesis methods in Tab. A6, and object insertion methods in Tab. A7.

## A.7 LLM Usage Disclosure

We used OpenAI's ChatGPT as a writing assistant during the preparation of this paper. Specifically, ChatGPT was employed for language refinement, grammar correction, and improving the clarity of exposition. It did not contribute to research ideation, experimental design, data analysis, or the derivation of technical results. All scientific content, including the conceptual framework, methodology, experiments, and conclusions, was created entirely by the authors.

Table A3: Training setup for the latent composition network $\Phi$.

| Item | Setting |
|------|---------|
| Datasets | Foreground from MAGICK dataset (Burgert et al., 2024) |
| | Background from BG20k dataset (Li et al., 2022) |
| Number of Samples | 20,000 |
| Latent Resolution | $64 \times 64$ |
| Optimizer | AdamW ($\beta_1 = 0.9, \beta_2 = 0.999$) |
| Learning Rate | $5 \times 10^{-6}$ |
| Weight Decay | $1e-2$ |
| Batch Size | 32 |
| LR Schedule | Cosine with restarts |
| Training Steps | $10,000$ |
| Gradient Clipping | 1.0 |
| Hardware | $1 \times$A100 80GB |
| Precision | Float32 |

Table A4: Quantitative evaluation of existing layered image synthesis methods on RORem dataset.

| | FG | Recomposite (or Background for LayeringDiff + ObjectClear) | | | |
|--------|----------|----------|----------|----------|----------|
| Method | CLIP ($\uparrow$) | MUSIQ ($\uparrow$) | MANIQA ($\uparrow$) | KID$\times$1000 ($\downarrow$) | DINO ($\uparrow$) |
| LayeringDiff + ObjectClear | 0.63 | 66.98 | 0.41 | 5.19 | **0.99** |
| LayerDiffuse | **0.77** | **70.05** | **0.43** | 9.04 | 0.50 |
| Ours | 0.70 | 68.34 | **0.43** | **4.73** | 0.94 |

Table A5: Quantitative evaluation with an object insertion method FluxFill and an image inpainting method PaintByInpaint on RORem datset.

| Method | MUSIQ ($\uparrow$) | MANIQA ($\uparrow$) | KID$\times$1000 ($\downarrow$) | DINO ($\uparrow$) |
|--------|----------|----------|----------|----------|
| PaintByInpaint (Wasserman et al., 2025) | 68.08 | 0.38 | 4.60 | 0.93 |
| FluxFill (Labs, 2024) | 67.72 | 0.42 | **4.59** | 0.93 |
| Ours | **68.34** | **0.43** | 4.73 | **0.94** |

Table A6: Quantitative evaluation of existing layered image synthesis methods on AnyInsertion dataset.

| | FG | Recomposite (or Background for LayeringDiff + ObjectClear) | | | |
|--------|----------|----------|----------|----------|----------|
| Method | CLIP ($\uparrow$) | MUSIQ ($\uparrow$) | MANIQA ($\uparrow$) | KID$\times$1000 ($\downarrow$) | DINO ($\uparrow$) |
| LayeringDiff + ObjectClear | 0.70 | 54.18 | 0.35 | 10.47 | **0.71** |
| LayerDiffuse | **0.73** | **65.36** | **0.40** | 8.51 | 0.41 |
| Ours | 0.71 | 55.54 | 0.32 | **3.88** | 0.45 |

Table A7: Quantitative evaluation with an object insertion method FluxFill and an image inpainting method PaintByInpaint on AnyInsertion datset.

| Method | MUSIQ (↑) | MANIQA (↑) | KID×1000 (↓) | DINO (↑) |
|---|---|---|---|---|
| PaintByInpaint (Wasserman et al., 2025) | **58.10** | **0.33** | 5.49 | 0.49 |
| FluxFill (Labs, 2024) | 55.01 | **0.33** | **3.02** | **0.56** |
| Ours | 55.54 | 0.32 | 3.88 | 0.45 |

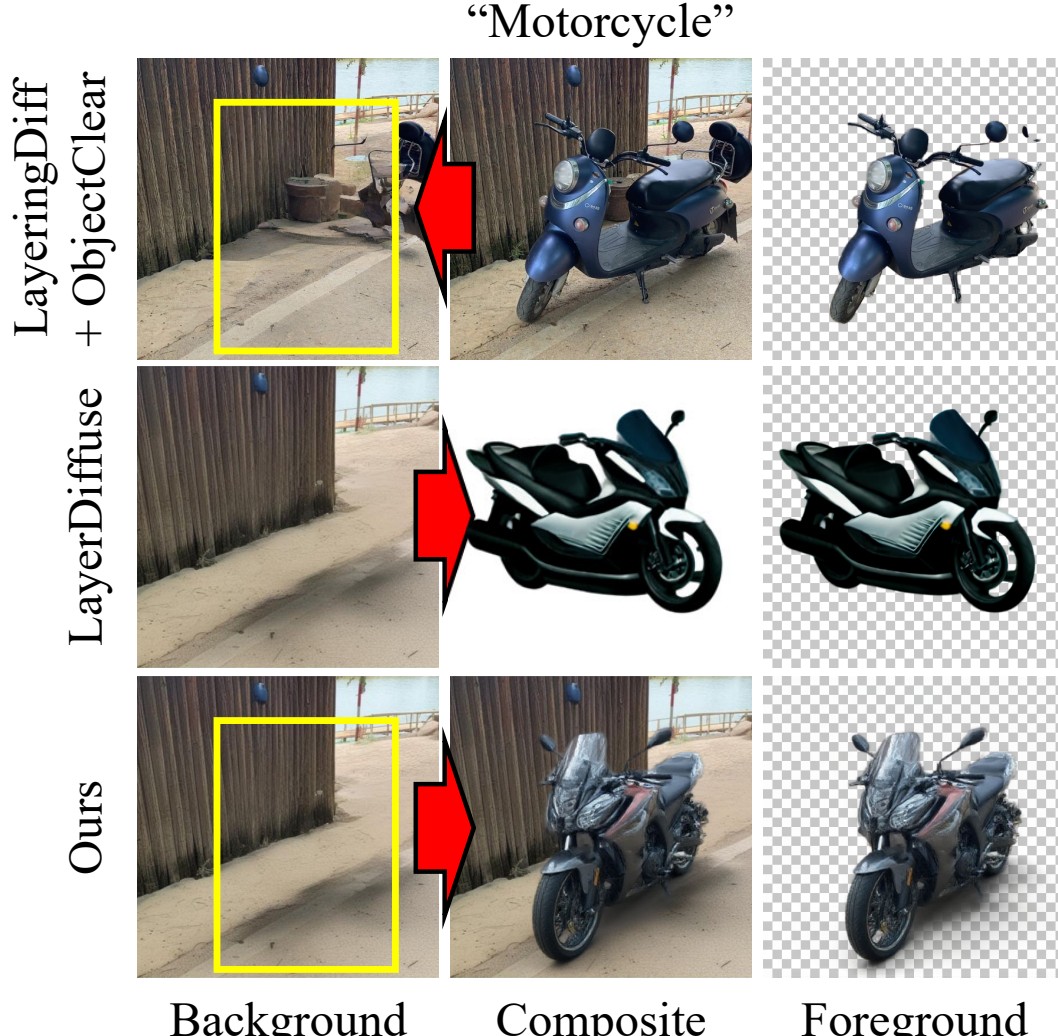

Figure A2: Qualitative comparison with existing layered image synthesis approaches.

"The new opel astra hatchback is shown in yellow"

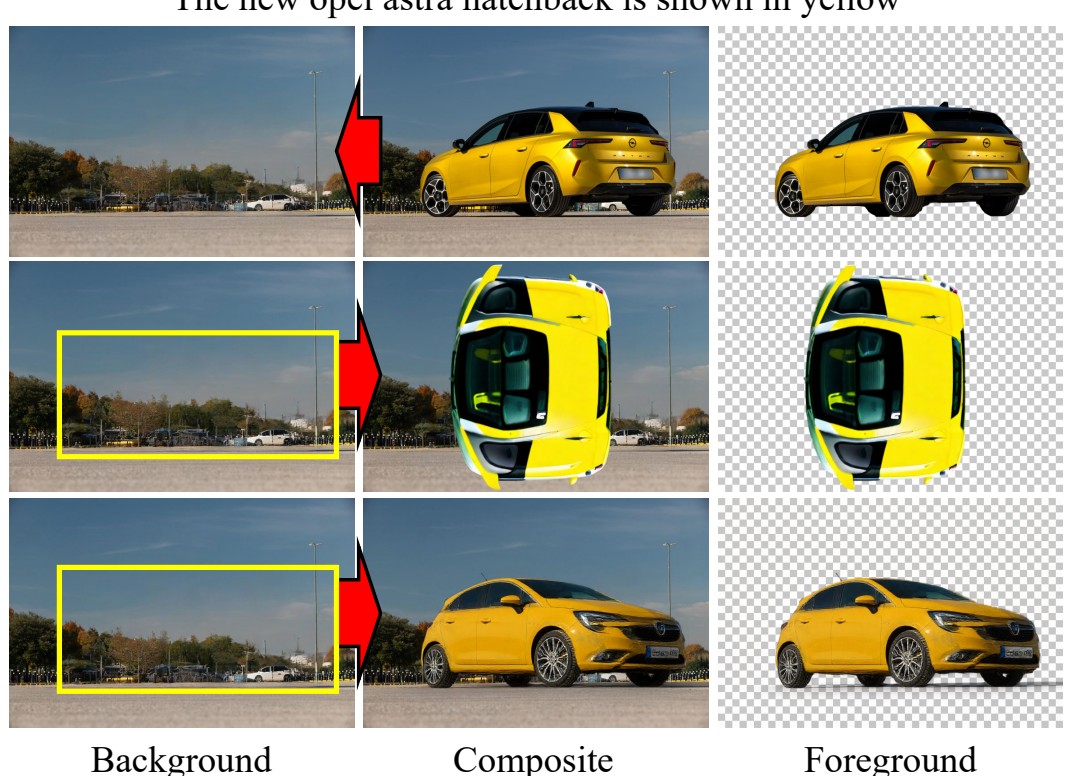

Background                    Composite                    Foreground

Figure A3: Qualitative comparison with existing layered image synthesis approaches.

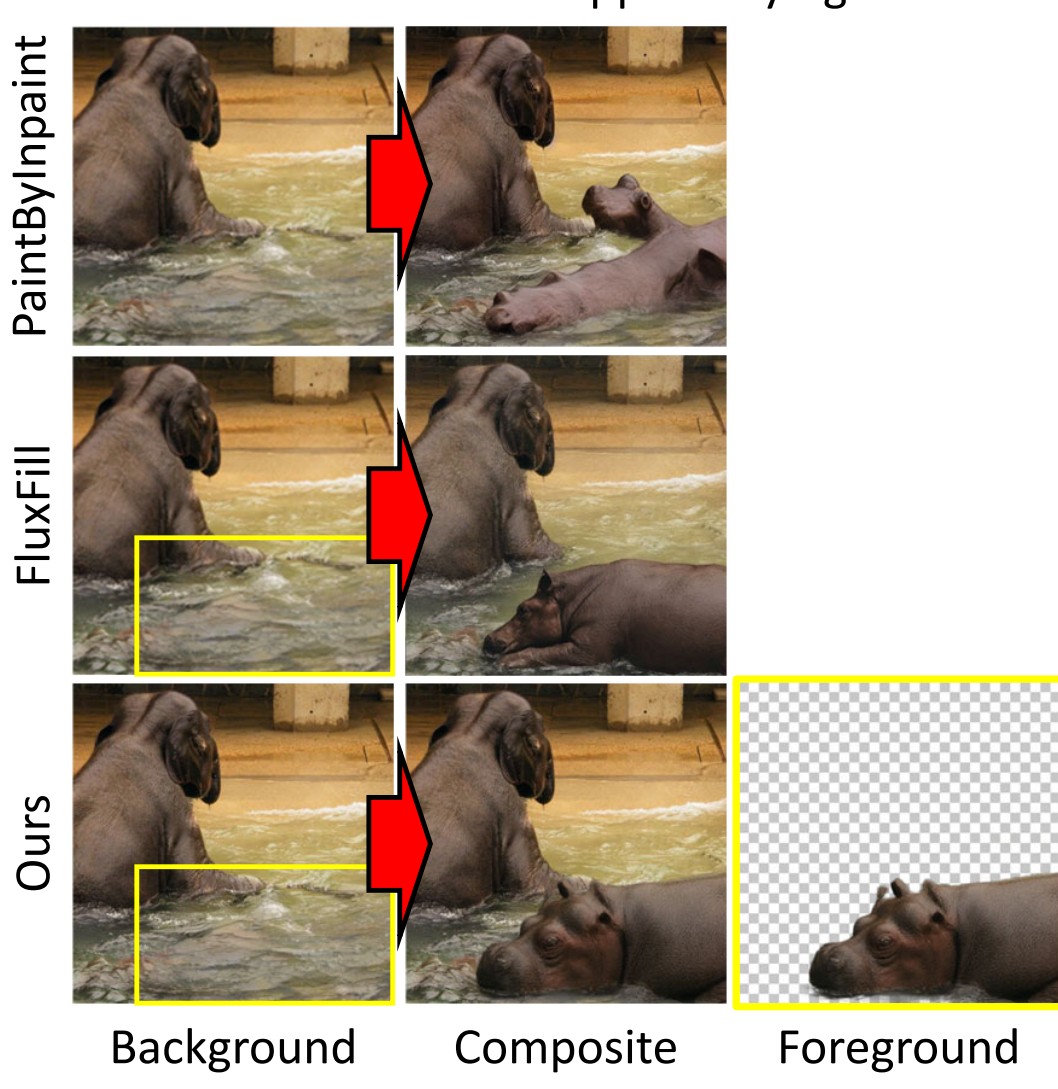

Figure A4: Qualitative comparison with an object insertion method PaintByInpaint (Wasserman et al., 2025) and a image inpainting method FluxFill (Labs, 2024)

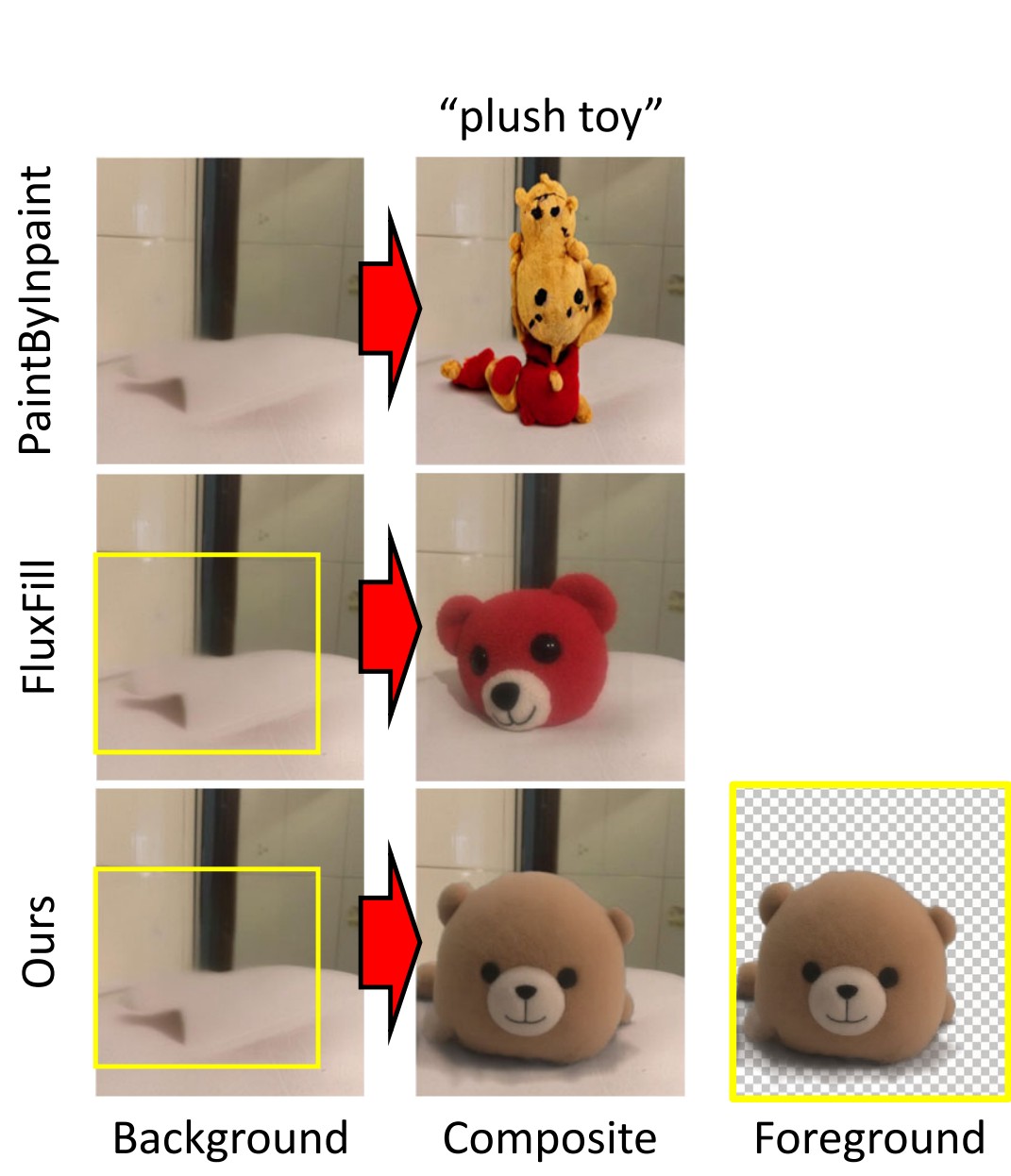

Figure A5: Qualitative comparison with an object insertion method PaintByInpaint (Wasserman et al., 2025) and a image inpainting method FluxFill (Labs, 2024)

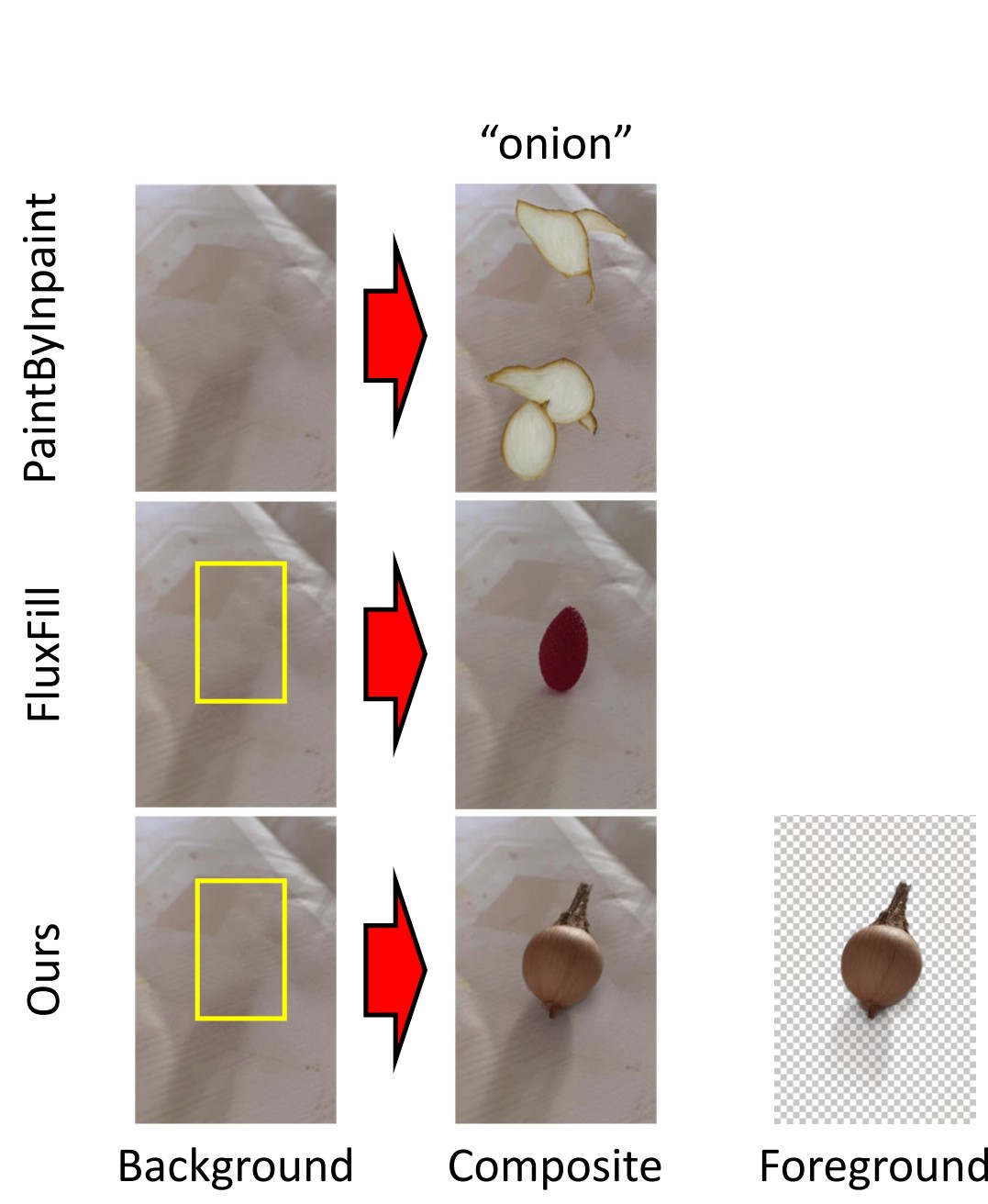

Figure A6: Qualitative comparison with an object insertion method PaintByInpaint (Wasserman et al., 2025) and a image inpainting method FluxFill (Labs, 2024)

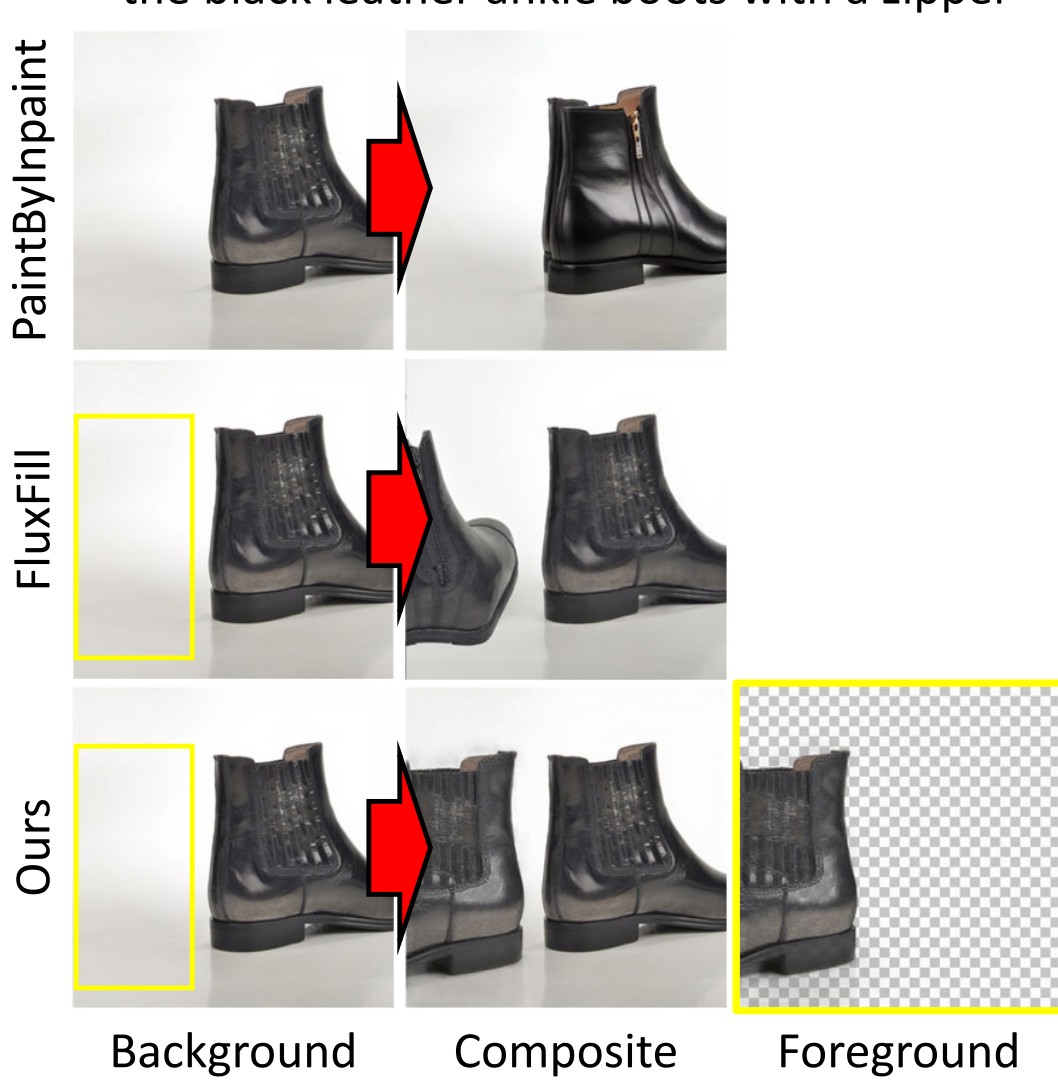

Figure A7: Qualitative comparison with an object insertion method PaintByInpaint (Wasserman et al., 2025) and a image inpainting method FluxFill (Labs, 2024)

Figure A8: Qualitative comparison with an object insertion method PaintByInpaint (Wasserman et al., 2025) and a image inpainting method FluxFill (Labs, 2024)

