# OpenReview forum: "BFS: BACK-TO-FRONT LAYERED IMAGE SYNTHESIS VIA KNOWLEDGE TRANSFER"
_ICLR.cc/2026/Conference — ICLR 2026 Conference Withdrawn Submission_

### Official Review · Reviewer_J6eb · 2025-10-30

**Soundness:** 2
**Presentation:** 2
**Contribution:** 1
**Rating:** 2
**Confidence:** 4

**Summary:**

This paper proposes BFS (Back-to-Front layered image Synthesis), a framework for layered image synthesis focusing on the 'background-to-foreground' (BG2FG) task. The goal is to generate an RGBA foreground layer that incorporates visual effects (e.g., shadows) and harmonizes with the background, given a background image and user guidance (text, mask). The paper proposes a 'knowledge transfer' strategy, attempting to use knowledge from 'composite image' synthesis to guide the synthesis of the 'foreground layer'. The method relies on a dual-branch architecture (generating a composite image and a foreground layer, respectively) built upon a shared pretrained diffusion Transformer (Flux-Fill) and connected by an information-sharing module.

**Strengths:**

- Facing the core bottleneck of the field—the difficulty of obtaining high-quality RGBA foreground layers with realistic visual effects—the authors leverage the relatively easier-to-learn knowledge of 'composite image' generation to guide the more challenging 'foreground layer' synthesis. This is a clever design to address data scarcity in layered generation.

- The paper designs an innovative dual-branch architecture. The key innovation is the use of a symmetric cross-attention 'information-sharing module' to couple the two branches, enabling bidirectional information exchange between the two generation pathways. This allows knowledge learned from the composite image branch (such as realism) to be effectively transferred to the foreground branch.

- Another major innovation is the two-stage training strategy that does not require ground-truth foreground layers. It uses simulated synthetic data to learn compositional relationships between foreground and background in the first stage, and then uses a dataset containing only background and real composite images to enhance realism in the second stage.

**Weaknesses:**

- The paper uses a backbone network (Flux-Fill) designed specifically for image inpainting to perform an RGBA generation task. Inpainting models are trained to preserve the background outside the mask, whereas RGBA models are trained to discard the background outside the object (i.e., generate transparency). The paper does not design any specific method or strategy to bridge this gap, and merely using LoRA for fine-tuning makes the entire method's foundation very unstable.

- The pipeline's effectiveness is highly dependent on the quality of the simulated data from Stage 1. However, the paper lacks a sensitivity analysis on this point. The example in Appendix Figure A1 (the teddy bear) itself shows that the simulated shadow is inconsistent with the scene's light source. The paper claims Stage 2 can 'enhance realism,' but this is a vague definition. If the simulation effect in Stage 1 is extremely poor (e.g., completely wrong shadow direction), it is impossible for Stage 2 to correct this from scratch.

- The experimental validation is very weak. The authors claim to surpass previous methods but only compare against LayerDiffuse among generative models. Using 'lack of public code' as an excuse for not comparing with other SOTA layered models like Text2Layer, DreamLayer, and ART is unacceptable. To prove the method's effectiveness, the authors should at least compare against a stronger baseline constructed by combining SOTA object insertion models and SOTA RGBA generation (matting) models.

**Questions:**

1. Please provide theoretical and experimental evidence to prove that an inpainting model (Flux-Fill) can be effectively converted into an RGBA generator using only LoRA fine-tuning. How does the model overcome its strong prior of identity mapping outside the mask and instead learn to output Alpha=0? For example, please show progressive visual results of the RGBA output during training.

2. If the quality of the simulated data in Stage 1 is very poor (e.g., completely wrong shadow direction, worse than Figure A1), will the training in Stage 2 fail? Or will it just learn to 'paste' this erroneous layer? Please provide a relevant sensitivity analysis and more results from your method showing shadow generation under significant lighting conditions to demonstrate the method's robustness.

3. In Stage 2, since there is no ground-truth foreground F, $\overline{F}=C \cdot M$ is used as a surrogate input for the foreground branch. This is a physically non-existent, 'dirty' input that mixes foreground and background. Why does this input not contaminate the (simulated) RGBA distribution that the foreground branch learned in Stage 1?

---

### Official Review · Reviewer_MRdd · 2025-10-30

**Soundness:** 2
**Presentation:** 2
**Contribution:** 2
**Rating:** 4
**Confidence:** 4

**Summary:**

The paper presents Back-to-Front layered image Synthesis, BFS, a generation-based framework for layered image synthesis aimed at addressing the limitations of existing decomposition-based (poor layer separation in complex scenes) and generation-based (difficult training data construction, suboptimal quality) methods. BFS adopts a background-to-foreground (BG2FG) approach, using a dual-branch framework to jointly generate composite images and foreground layers (enabling bidirectional information exchange for knowledge transfer) and a two-stage training scheme that avoids reliance on hard-to-obtain ground-truth foreground layers. Extensive experiments on datasets like SAM-FB, RORem, and AnyInsertion show BFS outperforms baselines such as LayeringDiff + ObjectClear and LayerDiffuse, producing high-quality layered images with well-harmonized foreground and background; it also supports practical applications like reference-based foreground generation and foreground layer extraction.

**Strengths:**

- Good Synthesis Quality and Layer Harmony: Compared with baseline methods such as LayeringDiff (decomposition-based) and LayerDiffuse (generation-based), BFS can generate foreground layers with visual effects like shadows.

- Practical Expansion Capability: BFS not only realizes basic layered image synthesis but also supports two practical applications: reference-guided foreground generation and foreground layer extraction, which expands the application scenarios of layered image technology.

**Weaknesses:**

- Missing visualization for multi-layer image generation. The paper mainly compares against LayerDiffuse. It would be beneficial to explicitly visualize whether the proposed method can extend two-layer synthesis to multi-layer image generation by continuously adding new foreground layers to the composited multi-layer image. In this process, how does the method handle object overlap across different layers? Additionally, will the model adjust the pose or motion of objects in the foreground layer to suit the background scene?

-  Missing hyperparameter details and weight initialization information. The paper fails to disclose hyperparameter specifics for the LoRA layers and Info Sharing module. Furthermore, does the model adopt the pretrained weights of LayerDiffuse for initialization?

- Missing ablation study on the necessity of output type. Given that the background image, generated foreground RGBA images, and corresponding masks are already available, why is it necessary to explicitly generate a composite image? The composite result can be directly derived by compositing the background and foreground via the mask.

**Questions:**

How is the model's performance in generating semi-transparent glass objects—specifically regarding the visibility of objects behind the glass—assessed in the paper? For instance, when generating glass with varying transparency or thickness, can the model accurately preserve the contour integrity and detail clarity of background objects seen through the glass? Does it avoid distortions (e.g., blurring, misalignment) caused by light refraction while ensuring the glass itself maintains a realistic semi-transparent appearance?

---

### Official Review · Reviewer_G5Hx · 2025-11-01

**Soundness:** 3
**Presentation:** 2
**Contribution:** 2
**Rating:** 4
**Confidence:** 4

**Summary:**

The paper introduces BFS, Back-to-Front layer synthesis, which is a method that builds on top of Flux-Fill. Focusing on the problems of the dependency on attention maps for multi-layer synthesis, and the difficulty of obtaining a high quality dataset, the authors tackles layered synthesis with a DiT-based approach, where the foreground generation branch and composite generation branch are designed in a way that they engage in information exchange. Starting from a given background image, the authors design their module that both a composite image (BG + FG) and the requested foreground layer is generated. To tackle the problem of the difficulty of obtaining a high quality dataset, the authors initially perform one training with synthetic data, where the model is fine-tuned with high-quality and realistic samples in the second stage. As the training objective, the authors introduce a latent level loss depending on the 1-step estimate of the denoising DiT, in addition to flow-matching loss which involves a regularization term for the second stage of training.

**Strengths:**

- The paper introduces a 2-stage training strategy for layered image synthesis, which can effectively handle scenarios that are hard to find training data.
- Compared to existing layered generation baselines, that mainly rely on U-Net based architectures, BFS reports noticeable quality improvements.
- Authors provide sufficient ablations to demonstrate the effectiveness of each added component, including the loss terms and the training stages.
- While not being the best in prompt alignment (which is an acceptable trade-off, given the qualitative results), the proposed method outperforms the competing approaches in recomposition benchmarks.

**Weaknesses:**

- My main concern regarding this paper is the presented use cases. Given the applicability of the layered images, there are two main advantages of such approaches. First is, with layers, they enable spatial editing trivial. For the second use case, the transparency property of RGBA images enables the synthesis of such objects. Given the examples in the paper, we have no clue on whether these cases are possible with the given pipeline.
- Over the given examples, the usability of the method is questionable. The authors provide certain use cases in Fig. 6, but given how usable the generated layers are, it is not clear how useful these cases are, compared to the baseline inpainting model other than improvements in terms of the quality. If the goal is to obtain a data generation pipeline, presenting the results in the form of a dataset may be more useful for this work.
- While the authors qualitatively provide ablations on training stages, since the goal is the quality of the generated layers, providing quantitative experiments or additional qualitative examples would be better. Otherwise, provided examples may be caused by biased examples.
- The authors enable information sharing in the single-stream blocks of Flux-Fill. Is there a reason of this choice? It would be a good practice if the authors can provide related ablations (between single-stream and double-stream blocks).

**Questions:**

- Can the method handle multi-foreground scenarios? Is iterative use of the method possible for generating a scene with multiple foregrounds?
- Can the method handle the generation transparent objects? This is a crucial property for the layered synthesis task.
- Given the generated foreground, background and a shifted mask, does the method enable spatial editing?

---

### Official Review · Reviewer_JXpS · 2025-11-02

**Soundness:** 3
**Presentation:** 2
**Contribution:** 2
**Rating:** 6
**Confidence:** 4

**Summary:**

This paper tackles generating images with layers. Layers are useful for editing.

Problem statement:
* Decomposition-based methods are prone to wrong separation of layers
* Generation-based methods lack compositional data to train models.

Solution:
* a background layer + user guidance -> foreground layer
* Foregrounds with shadow and reflection are coherent to the background.
* Dual-branch framework:
    * jointly generate a composite image and a foreground layer
    * bidirectional knowledge transfer between the branches
* Two-stage training strategy:
    * Jointly train the dual-branch model with limited dataset.
    * Fine-tune the composite branch with rich dataset.

Intuition:
* The compositional branch is easy to train.
* The foreground branch is stably trained with limited data because it is conditioned on the composite branch.

**Strengths:**

Originality:
1. The proposed method is new regarding the architecture and the training scheme.

Quality:
1. Just okay-ish.

Clarity:
1. The explanations are kind to the readers, step-by-step from the compositional image to training strategy.

Significance:
1. The model is light because the two branches are built with two LoRAs
2. The proposed dataset is useful: precise pixel-level compositional images.

**Weaknesses:**

1. The contribution of this paper should be clarified by comparison to the previous papers. Related work should discuss why the proposed method is special compared to previous approaches. L152 helps a bit. The layered generation is not new [Text2Layer]. Attention sharing  between from one process to another is not new [Masactrl, StyleAligned, StyleKeeper].
2. The paper does not accompany supplementary materials with many results. They will support that the results are not cherry-picked.


minor
1. Missing citation: FurryGAN: High Quality Foreground-aware Image Synthesis
2. “as discussed in (a previous section)” is redundant.

**Questions:**

Questions are apparent from the weaknesses. Addressing them may raise my rating.

---

### Note · Authors · 2025-11-12

**Comment:**

We sincerely thank the reviewers for their valuable and constructive feedback.
We have decided to withdraw this submission for now and plan to further improve the quality of our work by incorporating the reviewers’ comments.

**Withdrawal Confirmation:**

I have read and agree with the venue's withdrawal policy on behalf of myself and my co-authors.